# P720R USP8 Mutation Is Associated with a Better Responsiveness to Pasireotide in ACTH-Secreting PitNETs

**DOI:** 10.3390/cancers14102455

**Published:** 2022-05-16

**Authors:** Donatella Treppiedi, Giusy Marra, Genesio Di Muro, Emanuela Esposito, Anna Maria Barbieri, Rosa Catalano, Federica Mangili, Francesca Bravi, Marco Locatelli, Andrea Gerardo Lania, Emanuele Ferrante, Rita Indirli, Emma Nozza, Federico Arlati, Anna Spada, Maura Arosio, Giovanna Mantovani, Erika Peverelli

**Affiliations:** 1Department of Clinical Sciences and Community Health, University of Milan, 20122 Milan, Italy; donatella.treppiedi@unimi.it (D.T.); giusy.marra@unimi.it (G.M.); genesio.dimuro@unimi.it (G.D.M.); emanuela.esposito1@unimi.it (E.E.); anna.barbieri@unimi.it (A.M.B.); rosa.catalano@unimi.it (R.C.); federica.mangili@unimi.it (F.M.); francesca.bravi@unimi.it (F.B.); rita.indirli@unimi.it (R.I.); emma.nozza@unimi.it (E.N.); federico.arlati@unimi.it (F.A.); anna.spada@unimi.it (A.S.); maura.arosio@unimi.it (M.A.); erika.peverelli@unimi.it (E.P.); 2PhD Program in Endocrinological Sciences, University Sapienza of Rome, 00185 Rome, Italy; 3PhD Program in Experimental Medicine, University of Milan, 20054 Milan, Italy; 4Neurosurgery Unit, Fondazione IRCCS Ca’ Granda Ospedale Maggiore Policlinico, 20122 Milan, Italy; marco.locatelli@unimi.it; 5Department of Pathophysiology and Transplantation, University of Milan, 20122 Milan, Italy; 6Endocrinology, Diabetology and Medical Andrology Unit, Humanitas Clinical and Research Center, IRCCS, 20089 Rozzano, Italy; andrea.lania@hunimed.eu; 7Department of Biomedical Sciences, Humanitas University, 20090 Milan, Italy; 8Endocrinology Unit, Fondazione IRCCS Ca’ Granda Ospedale Maggiore Policlinico, 20122 Milan, Italy; emanuele.ferrante@policlinico.mi.it

**Keywords:** ACTH-secreting PitNETs, USP8, pasireotide, molecular marker

## Abstract

**Simple Summary:**

Somatic mutations in USP8 gene have been identified in 11–60% of patients with ACTH-secreting pituitary tumors. This study investigated the impact of USP8 mutations on corticotroph tumor cells responsiveness to the treatment with the somatostatin analog pasireotide. Although we found that the somatostatin receptor targeted by pasireotide, SSTR5, was upregulated in cells transfected with USP8 mutants, pasireotide failed to reduce ACTH secretion in cells expressing mutated USP8, except for the mutation P720R. Overall these data demonstrate that different USP8 mutations exert opposite effects on the responsiveness of ACTH-secreting pituitary tumors to pasireotide.

**Abstract:**

Somatic mutations in the ubiquitin specific peptidase 8 (USP8) gene have been associated with higher levels of somatostatin (SS) receptor subtype 5 (SSTR5) in adrenocorticotroph hormone (ACTH)-secreting pituitary neuroendocrine tumors (PitNETs). However, a correlation between the USP8 mutational status and favourable responses to pasireotide, the somatostatin multi-receptor ligand acting especially on SSTR5, has not been investigated yet. Here, we studied the impact of USP8 mutations on pasireotide responsiveness in human and murine corticotroph tumor cells. SSTR5 upregulation was observed in USP8 wild-type primary tumor cells transfected with S718del USP8 mutant. However, cell transfection with S718del USP8 and C40-USP8 mutants in in vitro sensitive cultures from USP8 wild-type tumors abolished their ability to respond to pasireotide and did not confer pasireotide responsiveness to the in vitro resistant culture. Pasireotide failed to reduce ACTH secretion in primary cells from one S718P USP8-mutated tumor but exerted a strong antisecretory effect in primary cells from one P720R USP8-mutated tumor. In agreement, AtT-20 cells transfection with USP8 mutants led to SSTR5 expression increase but pasireotide could reduce ACTH production and cyclin E expression in P720R USP8 overexpressing cells, only. In situ Proximity Ligation Assay and immunoflurescence experiments revealed that P720R USP8 mutant is still able to bind 14-3-3 proteins in AtT-20 cells, without affecting SSTR5 localization. In conclusion, P720R USP8 mutation might be considered as a molecular predictor of favourable response to pasireotide in corticotroph tumor cells.

## 1. Introduction

The discovery of recurrent mutations in the ubiquitin specific peptidase 8 (USP8) gene has represented a notable progress in the understanding of tumorigenesis of Cushing’s disease (CD) [1,2]. These mutations have been identified in 11–60% of patients with functioning corticotroph tumors [1,2,3,4,5,6,7,8,9,10,11,12,13,14,15] and are represented by missense variants or small single codon deletions that mostly occur within or in proximity to the consensus 14-3-3 binding motif RSYSSP at amino acid positions 715–720 [16,17,18,19,20]. In particular, the residues S718 and P720 are affected in 96.1% of all the samples carrying USP8 mutations [11]. The four mutations S718del, P720R, S718P and P720Q account for over 80% of the total number of genetic variants, S718del representing the most frequent mutation (25.3%), followed by P720R (24.9%) [11]. Some USP8 genetic variants are located outside the 14-3-3 binding motif, including the G664R mutation, located in the USP8 autoinhibitory region, described in one case [15]. Complex mutations affecting more than one amino acid have also been described, including always either S718 or P720 [11].

These changes lead to a dramatic loss of the physiological inhibitory role played by 14-3-3 proteins on USP8 deubiquitinase activity resulting in increased recycling of its substrates, as the case of the receptor tyrosine kinase epidermal growth factor receptor (EGFR) [1,2,15]. Because of sustained EGFR signalling, USP8 mutations promote cell proliferation and ACTH secretion [1,2].

The prognosis associated with genetic alterations in USP8 is still controversial, as USP8 mutated tumors seem to be associated with a greater likelihood of surgical remission [13,20] as well as a higher risk of recurrence of CD [15,21] compared to wild type tumors.

Of particular interest is that USP8 mutant tumors have been reported to have significantly higher mRNA and protein expression levels of somatostatin (SS) receptor subtype 5 (SSTR5) compared to wild type tumors [4,13] and a lower grade on clinicopathological classification, suggestive of a better prognosis. These findings have raised the question as to whether USP8 mutations are associated with drug susceptibility to the somatostatin analog (SSA) pasireotide.

Pasireotide, the somatostatin multi-receptor ligand acting especially on SSTR5, is the only approved pituitary tumor-targeting drug for the treatment of CD in adult patients [22]. However, its efficacy is variable and its administration may be associated with side-effects such as hyperglycemia, gall bladder disease and gastrointestinal symptoms [23]. Although an association between the expression of SSTR5 and response to this treatment has been suggested [24,25], predictive markers of responsiveness to pasireotide are lacking. At the same time, the value of USP8 mutational status as a marker of response to pasireotide still needs to be determined.

The aim of the present study was to evaluate the impact of USP8 mutations on SSTR5 expression and their correlation with responsiveness to pasireotide in primary cultures from surgically removed human corticotrophs tumors and murine corticotroph tumor cells AtT-20.

## 2. Materials and Methods

### 2.1. ACTH-Secreting Pituitary Cell Culture

The study was approved by the Fondazione IRCCS Ca’ Granda Ospedale Maggiore Policlinico Ethics Committee and patients with ACTH-secreting PitNETs who underwent trans-sphenoidal surgery gave their informed consent to the use of their sample. None of the patients was pharmacologically pre-treated with SSAs. Specifically, 7 ACTH-secreting PitNETs specimens (tumors #1–#7) were collected. All the tumors were non invasive microadenomas with low ki67 (<3%), except for tumor #6 that had a ki67 of 4%. Two tumors were not visualized at the MRI but only during surgery. Tumor samples were cut into two pieces: the first piece was immediately used for genomic DNA extraction and analysis on the mutational status of USP8, whilst the second piece was dissociated in order to obtain primary cell cultures, as described in previously published protocols [26]. Briefly, tumors were enzymatically dissociated with 2 mg/mL collagenase (Merck KGaA, Darmstadt, Germany) dissolved in Dulbecco’s modified Eagle’s medium (DMEM) at 37 °C for 2 h. Undigested material was removed by means of a 100-μm filter (nylon cell strainer, BD Transduction Laboratories, Lexington, UK). Dispersed cells were cultured in DMEM (Merck KGaA, Darmstadt, Germany) supplemented with 10% FBS, 2 mM glutamine and antibiotics (Gibco, Invitrogen, Life Technologies Inc., Carlsbad, CA, USA). In order to exclude fibroblast contamination, ACTH-secreting primary cultures were routinely checked by visual inspection with an optical microscope. Pituitary cells were kept at 37 °C in a humidified atmosphere with 5% CO_2_. When possible, primary cells from a single tumor were used for different types of in vitro experiments. Murine pituitary corticotroph tumor cell line AtT-20 (ATCC CRL-1795™) was kept in culture with DMEM (Life Technologies, Carlsbad, CA, USA) supplemented with 10% fetal bovine serum (FBS) and antibiotics (Life Technologies, Carlsbad, CA, USA). All experiments were performed with mycoplasma-free cells.

Pasireotide was provided by Novartis Pharma AG (Basel, Switzerland).

### 2.2. DNA Extraction and Sanger Sequencing

Genomic DNA was extracted from PitNETs specimens and AtT-20 cells with the Puregene Core Kit A following to the manufacturer’s instructions (Qiagen, Hilden, Germany). Genomic DNA from blood samples was isolated with the FlexiGene DNA Kit (Qiagen, Hilden, Germany). USP8 exon 14 was PCR amplified with GoTaq G2 DNA Polymerase (Promega, Madison, WI, USA) using a C1000 Touch Thermal Cycler (Bio-Rad Laboratories, Hercules, CA, USA) as previously described [15]. Direct sequencing was performed using the BigDye™ Terminator v3.1 Cycle Sequencing Kit and the 3130xl Genetic Analyzer (Applied Biosystems, Foster City, CA, USA).

### 2.3. Cell Transfection and USP8 Silencing

AtT-20 cells were seeded in 6-well plates at a density of 4 × 10^5^/well and transiently transfected for 96 h with pME-FLAG expression vectors coding for USP8 wild type and USP8 mutants (S718del, S718P, P720R and G664R, C40) generated as previously described [15]. Primary cultures were seeded in 24-well plates at a density of 12.5 × 10^4^/well. In case of USP8 wild-type tumors, primary cells were transiently transfected for 96 h with pME-FLAG expression vectors coding for S718del USP8 or USP8-C40. For both AtT-20 cells and primary cultures, empty vector was used as negative control in each experiment and Lipofectamine 2000 (Invitrogen, Thermo Fisher Scientific, Waltham, MA, USA) was used as transfection reagent. USP8 gene silencing was performed in AtT-20 cells using a murine SMARTpool of UPS8 predesigned small interfering RNAs (siRNAs) and Dharmafect transfection agent 4 (Dharmacon, GE Healthcare Life Sciences, Chicago, IL, USA) according to the instruction of the manufacturer. A negative control (scramble) siRNA was used in each experiment. The optimal concentration of siRNAs and the kinetics of silencing of USP8 were established in preliminary experiments and 72 h of silencing was chosen as optimal time point. For each experiment, transfection efficiency or USP8 silencing were checked by Western blot analysis using anti-FLAG or anti-USP8 antibodies (Santa Cruz Biotechnology, Dallas, TX, USA).

### 2.4. Measurement of ACTH Levels

After 72 h of transfection, AtT-20 cells were trypsinized, counted and re-seeded in a 24-well plate at a density of 12.5 × 10^4^ cells/well in 500 µL of culture medium at 37 °C. The following day cells were incubated with or without pasireotide 10 nM for 4 h [27]. Murine ACTH levels were determined in cell culture media using a specific Elisa immunoassay kit (Fine Test, Wuhan Fine Biotech Co., Ltd., Wuhan, China), according to the manufacturer’s instructions. Absorbance was read at 450 nm in a Victor2 multilabel plate reader (Perkin Elmer, Waltham, MA, USA). Data were plotted and analyzed with Curve Expert 1.4 program. Hormone levels were normalized on the protein content, measured by BCA assay. The assay was done in triplicate for each condition and experiments were replicated 3 times. Primary cells were transfected for 96 h and stimulated with or without pasireotide 10 nM for 4 h. Human ACTH in culture media was measured by specific chemiluminescent immunometric assay (Immulite 2000, Siemens Medical Solutions Diagnostics, Los Angeles, CA, USA) with an inter-assay coefficient of variation ranging from 6.1 to 10.0%, an intra-assay coefficient of variation ranging from 6.7 and 9.5% and sensitivity of 5 pg/mL.

### 2.5. Western Blot Analysis

30 µg of total proteins extracted from AtT-20 cells and primary cultured cells were separated on SDS/polyacrylamide gels and transferred to a nitrocellulose filter. USP8 and POMC antibodies were from Santa Cruz Biotechnology (Dallas, TX, USA) and diluted at 1:200; FLAG antibody was from Merck KGaA (Darmstadt, Germany) and diluted 1:1000; SSTR5 antibody was from Abcam (Cambridge, UK), cyclin E and cyclin D3 antibodies were from Cell Signaling Technology (Danvers, MA, USA) and were diluted 1:1000. Primary antibodies were incubated o/n at 4 °C. Anti-GAPDH antibody (Ambion, Thermo Fisher Scientific, Waltham, MA, USA) was used at 1:4000 for 1 h at room temperature. Secondary antibodies, anti-mouse or anti-rabbit (Cell Signaling Technology, Danvers, MA, USA) were incubated at room temperature for 1 h at 1:2000. Chemiluminescence was detected with ChemiDOC-IT Imaging System (UVP, Upland, CA, USA). Densitometrical analysis was performed with NIH ImageJ software. All of the original Western Blot figures shown in Appendix A, Appendix A, Appendix A, Appendix A, Appendix A and Appendix A.

### 2.6. Cell Proliferation Assay

Cell proliferation was assessed by colorimetric measurement of 5-bromo-2-deoxyuridine (BrdU) incorporation during DNA synthesis in proliferating cells for both AtT-20 cells and primary cultured cells (GE Healthcare, Life Science, Buckinghamshire, UK). 72 h transiently transfected AtT-20 cells were trypsinized, counted and re-seeded in starved medium in 96-well plate at a cell density of 5 × 10^5^ cells/well. USP8-mutated primary cells were seeded in starved medium in 96-well plate at a cell density of 5 × 10^5^ cells/well. The day after, starved medium of was replaced with complete medium containing or not pasireotide 10 nM for 96 h, based on preliminary time course experiments, for both AtT-20 and primary cultures. After that, BrdU incorporation was allowed for 2 h. The experiments were replicated 3 times. Each determination was done in triplicate.

### 2.7. Cell Apoptosis

At 72 h of transfection, AtT-20 cells were trypsinized, counted and re-seeded in complete medium in 96-well plate at a cell density of 2.5 × 10^4^ cells/well. The following day cells were stimulated or not with pasireotide 10 nM for 48 h at 37 °C, based on preliminary time course experiments. Cell apoptosis was measured using Apo-ONE homogenous caspase-3/7 assay (Promega, Madison, WI, USA). The liberation of Rhodamine 110 substrate was detected by an absorbance plate reader. Each determination was done in triplicate and experiments were repeated three times.

### 2.8. Fluorescence Microscopy and In Situ Proximity Ligation Assay

1.25 × 10^5^ cells/well AtT-20 cells were seeded on 13-mm poly-L-lysine coated coverslips in 24-well plates. After 48 h transfection, cells were fixed with 4% paraformaldehyde (Sigma-Aldrich, St. Louis, MO, USA) for 10 min at room temperature, washed three times with PBS and then incubated at room temperature with blocking buffer (5% FBS, 0.3% Triton™ X-100, in PBS) for 1 h. Cells were incubated overnight at 4 °C with anti-FLAG (1:250, Merck KGaA, Darmstadt, Germany), anti-SSTR5 (1:200, Proteinctech, Rosemont, IL, USA) or anti-pan 14-3-3 (1:200, Cell Signaling Technology, Danvers, MA, USA) antibodies. The anti-mouse Alexa Fluor™ -546-conjugated secondary antibody (1:500, ThermoFisher Scientific, Carlsbad, CA, USA) and anti-rabbit Alexa Fluor™ -488-conjugated secondary antibody (1:500, ThermoFisher Scientific, CA, USA) were incubated at room temperature for 1 h for immunofluorescence analysis. All antibodies were diluted in Antibody Diluent Reagent Solution (Life Techologies, ThermoFisher, Carlsbad, CA, USA). Negative control coverslips were incubated with only one primary antibody, respectively. In Situ Proximity Ligation Assay (PLA) (Duolink; Sigma-Aldrich, St. Louis, MO, USA) was performed as previously described [15]. Coverslips were mounted on glass slides with Duolink In Situ Mounting Medium with 4’,6-diamidino-2-phenylindole (DAPI) (Sigma-Aldrich, St. Louis, MO, USA). Proximity ligation events were quantified with NIH ImageJ software after image deconvolution as previously described [15].

### 2.9. Statistical Analysis

The results are expressed as the mean ± S.D. A paired two-tailed Student’s *t*-test was used to detect the significance between two series of data. *p* < 0.05 was accepted as statistically significant. In the case where the empty vector % secretion was 100% for all replicates the paired *t*-test was equivalent to a one-sample two-sided *t*-test that the average secretion for the mutant USP8 transfected cells was 100%.

## 3. Results

### 3.1. USP8 Mutants Upregulate SSTR5 Expression Levels in Primary Cultured Cells and AtT20 Cells

As initial step, we tested the impact of USP8 mutant on SSTR5 protein expression levels in cultured cells from three USP8-wild type ACTH-secreting PitNETs (#1, #2, #3) by S718del USP8 transient transfection. Indeed, S718del was chosen as one of the most representative USP8 mutant [2,11]. Western blot experiments showed SSTR5 upregulation in two out of three ACTH-secreting PitNETs upon S718del USP8 transient transfection (+1.20 and +2.55 SSTR5 fold-increase vs. empty vector for tumors #1 and #2, respectively) (Figure 1a). No changes in SSTR5 expression level was observed in the third tumor (#3).

SSTR5 expression was also investigated in murine corticotroph tumor AtT-20 cells, endogenously expressing USP8 wild-type. AtT-20 cells were transiently transfected with USP8 WT, S718P, S718del, P720R, G664R and C40. In particular, C40-USP8 was used as a catalytically active mutant whereas the recently identified G664R USP8 mutant located in the autoinhibitory region was included in this study for further investigation [15,28]. Western blot data showed a significant increase of SSTR5 protein expression in cells transfected with all USP8 mutants, with the exception of S718P USP8. P720R mutant showed the highest SSTR5 protein amount (+49.1 ± 21% vs. empty vector transfected cells, *p* < 0.01), (Figure 1b). USP8 transfection was evaluated by using a specific antibody recognizing both human and murine USP8, and, as expected, transfected human USP8 and endogenous murine USP8 were visualized as two separate bands due to differences in their molecular weight. In addition, AtT-20 cells transiently transfected with USP8 siRNA resulted in significant SSTR5 expression level decrease (−27.8 ± 1.6% reduction vs. siRNA C-, *p* < 0.01), (Figure 1c).

### 3.2. USP8 Mutants Dissimilarly Modulate the Sensitiveness to Pasireotide-Mediated Antisecretory Action in Primary Cultured Cells

The effect of USP8 mutations on tumoral corticotroph cells responsiveness to the antisecretory action of pasireotide was first evaluated in primary cultures from seven human ACTH-secreting PitNETs (five USP8 wild-type #1, #2, #3, #4, #5, respectively, and two USP8-mutated in S718P, #6, and P720R, #7, respectively). The mutations identified were not present in the germline. Four hours incubation with pasireotide resulted in ACTH secretion inhibition in 4 out of 5 primary cultures from different USP8 wild-type ACTH-secreting PitNETs. Two of the four in vitro sensitive tumors (−17.1 ± 5.7% and −29.3 ± 1.1% secretion vs. basal for tumors #1 and #2, respectively, *p* < 0.05) were transiently transfected with S718del USP8 mutant while the remaining two in vitro sensitive tumors (−38.2 ± 2% and −22.7 ± 3.5% secretion vs. basal for tumors #4 and #5, respectively, *p* < 0.05) with the C40-USP8 mutant. In all cases cell transfection with USP8 mutants abolished the ability of the tumors to respond to pasireotide (Figure 2a,b). Transient cell transfection with S718del USP8 mutant in the in vitro resistant tumor (#3) did not confer pasireotide responsiveness (Figure 2c). Accordingly, no pasireotide antisecretory effect was observed in primary cultured cells from one ACTH-secreting pituitary tumor endogenously carrying the S718P USP8 mutation (#6) (Figure 2d). On the contrary, a strong reduction of ACTH release was achieved in primary cultured cells from one ACTH-secreting pituitary tumor bearing the P720R USP8 mutation incubated with pasireotide (#7) (−46.4 ± 11% secretion vs. basal *p* < 0.05) (Figure 2e).

### 3.3. P720R USP8 Is the Sole Mutant Showing Favourable Responses to Pasireotide in AtT-20 Cells

We further investigated the involvement of USP8 mutations in SSTR5-mediated intracellular signalling in AtT-20 cells. To this purpose, we tested the ability of pasireotide to reduce POMC expression levels and ACTH secretion and to exert cytostatic and cytotoxic effects in AtT-20 cells transfected with USP8-mutants.

USP8 mutant transfection in AtT-20 cells induced an increase of ACTH secretion and cell proliferation in the absence of pasireotide incubation (data not shown), as previously reported [15]. Here, pasireotide failed to reduce POMC expression and ACTH secretion in AtT-20 cells transfected with WT USP8 and all USP8 mutants with the exception of P720R mutant. Similar to empty vector transfected cells, P720R USP8 expressing cells were responsive to pasireotide in terms of reduction of POMC expression (−18.9 ± 15.5%, *p* < 0.05 vs. basal) and ACTH release (−39.2 ± 19.3%, *p* < 0.01 vs. basal) (Figure 3a,b).

Moreover, as in empty vector transfected cells, pasireotide incubation resulted in a significant decrease of cyclin E levels in P720R USP8 mutant expressing cells, only (−21.4 ± 14.1%, *p* < 0.05 vs. basal) (Figure 3c). However, this effect was not associated to a reduction of the overall proliferation rate since upon pasireotide incubation, reduction in cell proliferation (−22.8 ± 3.7% vs. basal, *p* < 0.001) was observed in empty vector transfected cells, only (Figure 3d). Similarly, an increase in cell apoptosis (+42.2 ± 31% vs. basal, *p* < 0.05) was observed in empty vector transfected cells but not in USP8-transfected cells (Figure 3e).

To better characterize the impact of USP8 mutations on pasireotide-mediated regulation of cell cycle, we tested cyclin D3 expression in primary cultured cells from two wild-type USP8 tumors (#1 and #2) and one P720R USP8-mutated tumor (#7) exposed to pasireotide for 4 h. Pasireotide failed to reduce cyclin D3 expression in two USP8 wild-type primary cultures tested and cell transfection with S718del USP8 mutant did not confer pasireotide responsiveness (Figure 4a). Likewise, pasireotide could not reduce cyclin D3 expression nor cell proliferation in the P720R USP8-mutated primary culture (Figure 4b).

### 3.4. P720R USP8 Mutant Exhibits 14-3-3 Proteins Binding Ability in AtT-20 Cells

In order to elucidate possible molecular mechanisms determining the favourable responsiveness to pasireotide-mediated antisecretory actions in P720R USP8 expressing cells, we tested the ability of this mutant to influence SSTR5 localization and bind 14-3-3 proteins in AtT-20 cells. As shown by immunofluorescence images, neither P720R USP8 nor S718del USP8 affected SSTR5 localization (Figure 5a). Indeed, as in USP8 wild type cells, SSTR5 was mainly located at the plasma membrane under basal condition.

Interestingly, PLA experiments demonstrated that, contrary to S718del USP8 mutant, the P720R USP8 mutant maintained the capability to correctly bind 14-3-3 proteins (Figure 5b). NetPhos 2.0 server indicated that the amino acid substitution of proline 720 to arginine does not affect the probability of serine 718 to be potentially phosphorylated (score 0.997 in P720R USP8 vs. 0.992 in wild-type USP8).

### 3.5. The Loss of Pasireotide Responsiveness in USP8 Wild Type Overexpressing AtT-20 Cells Is Due to Abundant USP8 Levels

Finally, we attempted to understand whether the lack of responsiveness to pasireotide-mediated antisecretory effects observed in USP8-wild type overexpressing cells was due to an excess of USP8 molecules within the cells. To this purpose, AtT-20 cells were transiently transfected with different amounts of plasmid encoding the USP8 wild type form. According to the plasmid concentration curve shown in Figure 6a, a USP8 wild type plasmid DNA amount of 0.75 μg was chosen as the lowest one resembling those of endogenous USP8 levels. Cells were exposed to pasireotide for 4 h and POMC protein expression levels was evaluated by Western blot. Under these transfection conditions, similarly to what observed in empty vector cells, USP8-transfected cells incubation with pasireotide resulted in a significant reduction of POMC (−13 ± 4% reduction vs. basal, *p* < 0.05).

## 4. Discussion

Pasireotide is the multi-somatostatin receptor ligand with higher affinity for SSTR5 currently used to pharmacologically treat pituitary corticotroph tumors. However, biochemical remission is observed only in a subset of cases and molecular predictors of responsiveness to pasireotide are still lacking [29]. In this scenario, the present study aimed to determine the impact of USP8 mutations on in vitro responsiveness to pasireotide both in primary cultured cells from ACTH-secreting PitNETs and murine corticotroph tumor AtT-20 cells, in order to assess whether they might be considered predictable markers of favorable drug response.

First of all, in line with previously published data showing higher SSTR5 expression level in corticotroph tumor tissues bearing USP8 mutations [4,13], we also observed SSTR5 upregulation in primary cultured cells from USP8 wild-type ACTH-secreting PitNETs upon transient transfection with S718del USP8 mutant. Moreover, we reported a significant increase in SSTR5 protein expression in AtT-20 cells transfected with the most frequently encountered USP8 mutants, with the exception of S718P USP8, as well as with the recently identified G664R USP8 mutant located in the USP8 autoinhibitory region [15,28]. Consistent with these results, SSTR5 expression decrease was observed in USP8 silenced AtT-20 cells. To date, information regarding the mechanisms of transcriptional regulation for the SSTR5 gene is limited and whether ubiquitinated SSTR5 is a direct substrate of USP8 still remains to be assessed. Indeed, little is known about ubiquitination processes regulating endosomal sorting of SSTRs overall. Although it has been demonstrated that, contrary to murine sst2A, sst3 is subjected to ubiquitination-dependent lysosomal degradation in HEK293 transfected cells [30], no data about sst5 are available. Our data suggested that the elevated deubiquitinase (DUB) activity of the USP8 mutants might prevent SSTR5 downregulation, but further studies are required to better characterize the molecular mechanisms involved.

We then evaluated the impact of USP8 mutations on the in vitro responsiveness to pasireotide in corticotroph tumor cells. Cell transfection with S718del USP8 and C40-USP8 mutants in responsive primary cultures bearing wild-type USP8 caused a complete loss of their ability to respond to the antisecretory action of pasireotide. At the same time, cell transfection with S718del USP8 did not confer sensitivity to pasireotide to the in vitro resistant tumor. Accordingly, pasireotide failed to reduce ACTH secretion in the S718P USP8-mutated tumor cells but achieved a strong ACTH reduction in P720R USP8 mutated tumor cells. It is valuable noting that such ACTH reduction was even more pronounced than what observed in USP8 wild-type primary cultures. To the best of our knowledge, these data represent the first evidence showing that, although associated with SSTR5 upregulation, mutations occurring at the amino acid 718 of USP8 are not associated to a favorable response to pasireotide, whereas USP8 mutations occurring at the amino acid 720 might preserve pasireotide responsiveness. Although the availability of a small number of primary cultures and the use of only one P720R USP8-mutated tumor are major limitations of our study, data obtained in AtT-20 cells are strongly encouraging and seem to support these evidences. Indeed, experiments in AtT-20 cells confirmed that the P720R USP8 mutant represents the sole mutant able to maintain responsiveness to antisecretory effects and inhibitory actions on cyclin E of pasireotide, whereas the other USP8 mutants completely reverted not only cells sensitivity to pasireotide antisecretory effects but also to its antiproliferative and cytotoxic actions. However, when further analysing the antiproliferative potential of pasireotide in primary cultured cells transfected with S718del USP8 mutant or naturally bearing P720R USP8 mutation, we could not observe any inhibitory effects on cyclin D3 expression levels or cell proliferation.

Nevertheless, these results prompted us to further investigate the possible different molecular mechanisms underlying the ability of P720R USP8 and S718del USP8 mutants to respond or not to pasireotide antisecretory effects, respectively. To this purpose we evaluated the impact of P720R and S718del USP8 mutations on SSTR5 intracellular localization. As shown by our immunofluorescence analysis, no difference in the subcellular localization of SSTR5 was found, suggesting that the intracellular sorting of SSTR5 is not affected by these USP8 mutations. Consistent with this result, similar membrane expression patterns of SSTR5 have been previously reported among differently USP8 mutated tumors and between USP8-mutated and wild type tumors by IHC analysis [4]. Surprisingly, when analyzing the impact of P720R and S718del USP8 mutations on USP8 interaction with 14-3-3 proteins, we observed that P720R USP8 mutant was still capable to bind to 14-3-3 proteins, being this data in agreement with results obtained by means of fluorescence polarization assay by the group of Ottmann [18]. By NetPhos 2.0 server, we assessed that mutation of proline 720 to arginine does not affect the probability of serine 718 to be phosphorylated. This in silico evidence would explain the preserved binding between USP8 and 14-3-3 proteins observed in our experiments and support the crucial role of the phospho-site in driving the protein-protein interactions (PPIs), consistently with other previously studied 14-3-3 PPIs [31,32,33].

Ultimately, we observed a lack of pasireotide responsiveness in AtT-20 cells overexpressing the wild-type form of USP8. To this regard, it is possible to hypothesize that the excess of USP8 molecules within cells, without of a counterbalanced overexpression of 14-3-3 molecules, could have culminated in an unexpected and intensified USP8 DUB activity that have rendered the phenotype of these cells more similar to the ones observed in the majority of USP8 mutants transfected cells instead of the one of empty vector transfected control cells. Indeed, this resistant phenotype was reverted by lowering the amount of plasmidic DNA during transfection. In line with our observation, it has been previously observed that USP8-mutated tumors have a significantly higher USP8 mRNA and protein expression levels than the wild type USP8 tumors [2,4], however the molecular mechanisms underlying the upregulation of USP8 gene expression by its mutations remain unknown.

Altogether, our data demonstrate that, although USP8 mutations are associated with an increased SSTR5 expression, P720R USP8 represents the sole mutation that could be used as a marker to predict a favorable antisecretory response to pasireotide treatment in tumor corticotroph cells since other USP8 mutations have been shown to induce resistance to pasireotide antisecretory and anti-proliferative effects. Moreover, these findings offer valuable insights into the crucial USP8-14-3-3 interaction and might provide a starting point for future structure-based drug discovery studies on its modulation with the aim of targeting and stabilizing this PPI interface as a novel potential therapeutic strategy for CD.

## 5. Conclusions

The results of the present study show that USP8 mutations are associated with an increase of SSTR5 expression. However, the downstream effects depend on the USP8 residue involved. Indeed, mutations occurring at the amino acid 718 of USP8 are associated to resistance to pasireotide antisecretory and anti-proliferative effects, whereas P720R USP8 mutation preserves pasireotide responsiveness, suggesting that USP8 mutational status might represent a novel biomarker predicting pasireotide responsiveness in ACTH-secreting PitNETs.

## Figures and Tables

**Figure 1 cancers-14-02455-f001:**
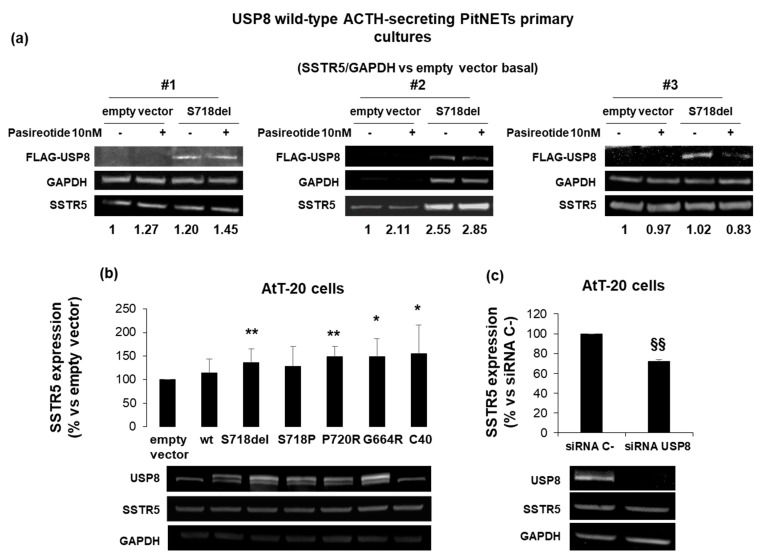
SSTR5 expression is upregulated by USP8 mutants and downregulated by USP8 genetic silencing. (**a**) Immunoblots of SSTR5 expression in three different primary cultures from USP8-wild type ACTH-secreting PitNETs (#1, #2, #3) transiently transfected with FLAG-S718del USP8 mutant and densitometrical analysis of SSTR5 expression normalized on GAPDH expression (single experiments). (**b**) Representative immunoblots of SSTR5 expression in AtT-20 cells transiently transfected with USP8 mutants and (**c**) AtT-20 cells silenced for USP8. The equal amount of protein was confirmed by stripping and reprobing with an anti-GAPDH antibody as housekeeping protein. The graphs show the quantification of SSTR5/GAPDH in all samples tested. *, *p* < 0.05, **, *p* < 0.01 vs. empty vector control cells; §§, *p* < 0.01 vs. SiRNA C-cells.

**Figure 2 cancers-14-02455-f002:**
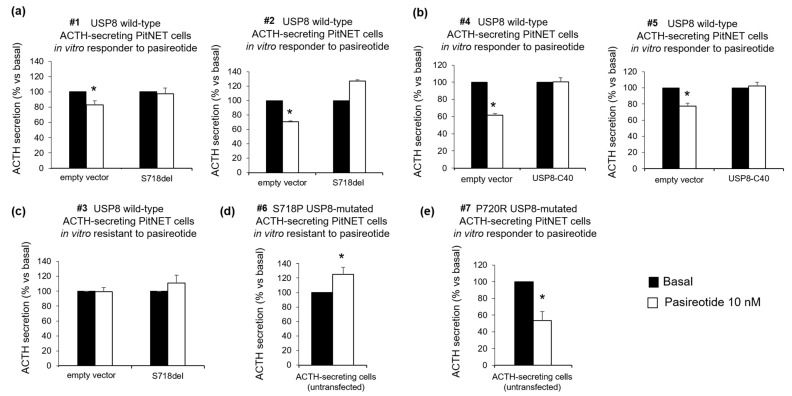
Effects of USP8 mutations on pasireotide-mediated inhibition of ACTH secretion in human primary cultures. Determination of ACTH release in (**a**) primary cultured cells in vitro responsive to pasireotide from two independent USP8 wild-type ACTH-secreting PitNETs (#1, #2) transiently transfected with S718del USP8 mutant, in (**b**) primary cultured cells in vitro responsive to pasireotide from two independent USP8 wild type ACTH-secreting PitNETs (#4, #5) transiently transfected with C40-USP8 mutant, in (**c**) primary cultured cells in vitro resistant to pasireotide from one USP8 wild-type ACTH-secreting PitNET (#3) transiently transfected with S718del USP8 mutant, (**d**) primary cultured cells from one S718P USP8 mutated ACTH-secreting PitNET (#6) and (**e**) primary cultured cells from one P720R USP8-mutated ACTH-secreting PitNET (#7). ACTH was measured in culture medium upon 4 h incubation with 10 nM pasireotide. No significant differences were found between basal ACTH secretion in empty vector and mutant USP8 transfected cells. Each determination was done in triplicate. Values represent mean ± SD and are expressed as percent of basal; *, *p* < 0.05 vs. corresponding basal.

**Figure 3 cancers-14-02455-f003:**
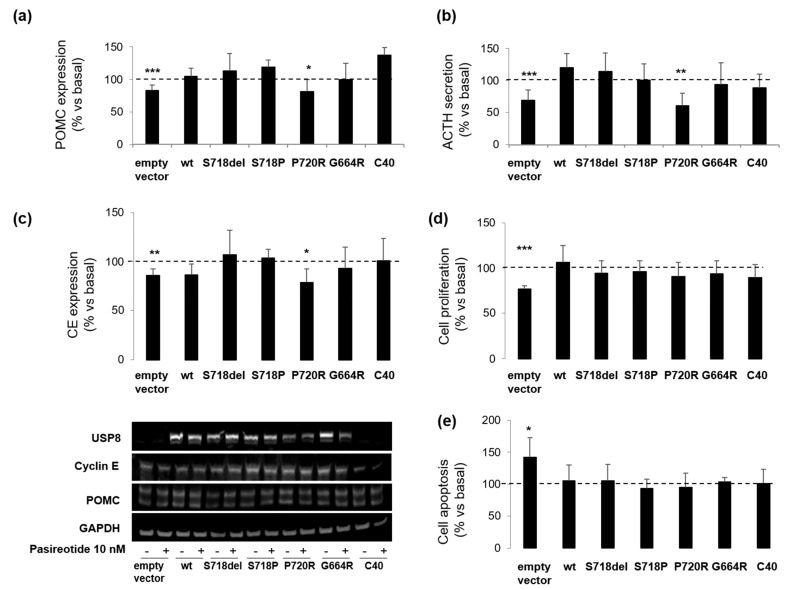
Effects of USP8 mutations on pasireotide-mediated signaling in AtT-20 cells. AtT-20 cells transiently overexpressing USP8 mutants were incubated with pasireotide 10 nM for specific times as indicated in M&M section. Representative immunoblots and corresponding graphs showing densitometrical analysis of (**a**) POMC and (**c**) cyclin E expression in AtT-20 cells. The equal amount of proteins was confirmed by stripping and reprobing with an anti-GAPDH antibody as housekeeping protein. Values represent mean ± SD of three independent experiments and are expressed as percent of basal untreated cells. *, *p* < 0.05, **, *p* < 0.01, ***, *p* < 0.001 vs. corresponding basal. Results of (**b**) ACTH determination, (**d**) cell proliferation assay and (**e**) caspase-3/7 assay in AtT-20 cells. Each type of experiment was repeated three times, and each determination was done at least in triplicate. Values represent mean ± SD and are expressed as percent respect to basal untreated cells *, *p* < 0.05, **, *p* < 0.01, ***, *p* < 0.001 vs. corresponding basal.

**Figure 4 cancers-14-02455-f004:**
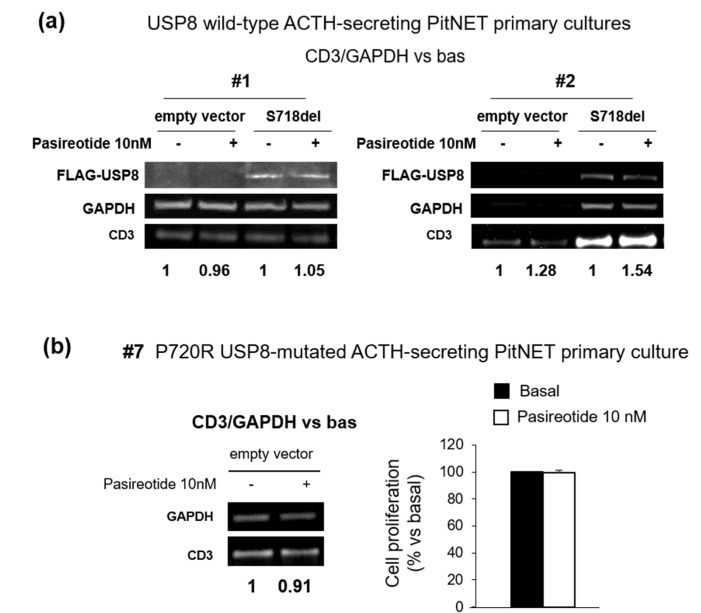
Evaluation of USP8 mutations on cell cycle regulation in human primary cultures. (**a**) Immunoblots and densitometrical analysis of CD3 expression in primary cultured cells from two independent USP8 wild-type ACTH-secreting PitNETs (#1 and #2) transiently transfected with S718del USP8 mutant for 96 h and incubated with pasireotide 10 nM for 4 h (single experiments). “Bas” is referred to cells incubated without pasireotide. (**b**) Immunoblot and densitometrical analysis of CD3 expression in primary cultured cells from one P720R USP8-mutated ACTH-secreting PitNET (#7) incubated with pasireotide for 4 h and results of cell proliferation assay obtained in cells treated with pasireotide for 96 h (single experiments). For western blot analysis, the equal amount of proteins was confirmed by stripping and reprobing with an anti-GAPDH antibody as housekeeping protein. The graphs show the quantification of CD3/GAPDH in all samples tested. For cell proliferation assay each determination was done in triplicate. Values represent mean ± SD and are expressed as percent respect to basal untreated cells.

**Figure 5 cancers-14-02455-f005:**
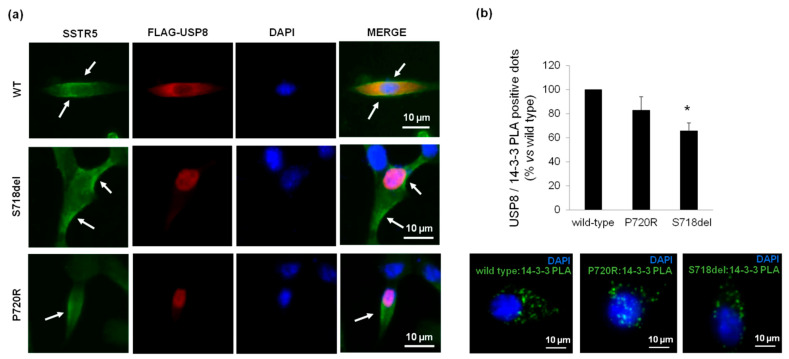
Impact of USP8 mutations on SSTR5 localization and USP8-14-3-3 proteins interaction. (**a**) Representative immunofluorescence experiment showing subcellular localization of SSTR5 in AtT-20 cells transiently transfected with plasmids encoding FLAG-tagged WT USP8 (upper panel), S718del USP8 (middle panel) or P720R USP8 (bottom panel). Cells were fixed and immunostained for endogenous SSTR5 (green) and FLAG-USP8 (red). Cell nuclei are stained in blue with DAPI. Arrows indicate major SSTR5 cell localization (plasma membrane) for each condition. For each condition, at least 10 cells from three independent experiments were analysed. Scale bars, 10 μm. (**b**) Representative In Situ PLA experiment performed in AtT-20 cells transfected with FLAG-tagged USP8 WT, S718del USP8 or P720R USP8. AtT-20 cells were incubated with anti-FLAG and anti-14-3-3 antibodies, respectively. Mouse anti-FLAG and rabbit anti-14-3-3 probes were used, respectively. Each picture represents a typical cell staining observed in 10 fields randomly chosen. Green dots represent PLA events and indicate close proximity between USP8 and 14-3-3 proteins. Cell nuclei are stained in blue with DAPI. Scale bars, 10 μm. The graph shows quantification of total USP8/14-3-3 puncta representing PLA events (3 independent experiments, number of PLA puncta per cell was quantified for 150 cells randomly chosen from different fields per condition, *, *p* < 0.05 vs. WT USP8 transfected cells).

**Figure 6 cancers-14-02455-f006:**
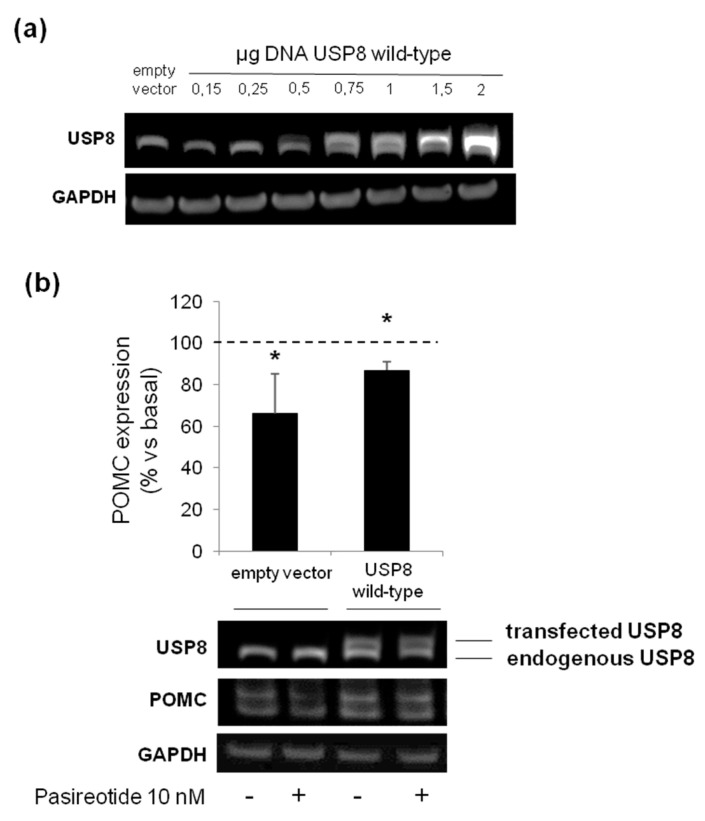
Evaluation of USP8 overexpression on AtT-20 cells responsiveness to pasireotide. (**a**) Immunoblot showing USP8 expression according to increasing concentration of DNA amount of plasmid encoding wild-type USP8 transfected in AtT-20 cells. Endogenous and transfected wild-type USP8 are both recognized by an anti-USP8 antibody as lower and upper bands, respectively. GAPDH bands are shown. (**b**) Representative immunoblot and densitometrical analysis of POMC expression normalized on GAPDH bands in AtT-20 cells transiently transfected with 0.75 µg DNA of wild type USP8 and treated with pasireotide 10 nM for 4 h. Values represent mean ± SD of three independent experiments and are expressed as percent of basal untreated cells. *, *p* < 0.05, vs. corresponding basal.

## Data Availability

Not applicable.

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
