# Peer review of "P720R USP8 Mutation Is Associated with a Better Responsiveness to Pasireotide in ACTH-Secreting PitNETs"

_cancers, 2022, doi:10.3390/cancers14102455_

Round 1
Reviewer 1 Report
The authors reported USP8 mutations on responsiveness to pasireotide both in primary cultured cells from ACTH-secreting PitNETs and murine corticotroph tumor AtT-20 cells. The results interfere a novel potential therapeutic strategy for Cushing Disease.
Below are my comments.
Title- Direct titles are easier to understand. I think the word "P720R" is necessary.
L212- As S718del is one of the most representative USP8 mutants, it is better to write an introduction about the percentage of other mutants (such as S718P, S718del, P720R, G664R and C40, and especially P720R) in corticotroph PitNET. Also, do multiple mutations be confirmed at the same time? For example, do S718del and P720R exist at the same time?
L231(Fig1)- The FLAG-USP lane contrast are too strong. Please display them properly. In addition, the font and size of characters are not unified in the figure. Also, the black boxes that indicate the band's lanes are not the same size. Error bars are mandatory even if standardized on 100% line.The same is true for other figures.
L262(Fig2)- The reaction of each pasireotide when transfecting empty vector and mutant vector is standardized and shown. Is there any result of direct comparison between empty vector and mutant vector?
L274- There are two periods at the end. The authors should also read the other texts carefully to see if there are any such mistakes. In addition, the authors should use the English proofreading service as needed.
L294(Fig3)- From (a) to (e), the characters on the horizontal axis are uneven and very difficult to see. Also, I'm worried that the letters of POMC are off center. Check all figures once for such mistakes.
L313(Fig4)- What was “bas” ? (a) The FLAG-USP and GAPDH photos are the same as in Fig. 1, so they should be taken for only Fig. 4.
Author Response
Reviewer 1
We thank the reviewer for the careful reading of our manuscript and for her/his comments. All the issues raised have been addressed.
- Title- Direct titles are easier to understand. I think the word "P720R" is necessary.
We have changed the title of the manuscript including P720R as follows: “P720R USP8 mutation is associated with a better responsiveness to pasireotide in ACTH-secreting PitNETs”.
L212- As S718del is one of the most representative USP8 mutants, it is better to write an introduction about the percentage of other mutants (such as S718P, S718del, P720R, G664R and C40, and especially P720R) in corticotroph PitNET. Also, do multiple mutations be confirmed at the same time? For example, do S718del and P720R exist at the same time?
As suggested by the Reviewer, we added in the revised manuscript an introduction about the frequency of the USP8 mutants and the multiple mutations (lines 67-74): “In particular, the residues S718 and P720 are affected in 96.1% of all the samples carrying USP8 mutations (11). The four mutations S718del, P720R, S718P and P720Q account for over 80% of the total number of genetic variants, S718del representing the most frequent mutation, followed by P720R (11). Some USP8 genetic variants are located outside the 14-3-3 binding motif, including the G664R mutation, located in the USP8 autoinhibitory region, described in one case (15). Complex mutations affecting more than one amino acid have also been described, including always either S718 or P720 (11).”
- L231(Fig1)- The FLAG-USP lane contrast are too strong. Please display them properly. In addition, the font and size of characters are not unified in the figure. Also, the black boxes that indicate the band's lanes are not the same size. Error bars are mandatory even if standardized on 100% line. The same is true for other figures.
We have replaced FLAG-USP8 pictures in Fig.1a and unified font and size of characters. The graph in fig.1b represents mean+-SD normalized vs empty vector. We deleted the 100% line that in this graph is redundant. In this figure and in the other figures, basal does not present error bar since is the mean of different experiments in each of them the values of the treated groups have been expressed as a percentage of the respective basal.
- L262(Fig2)- The reaction of each pasireotide when transfecting empty vector and mutant vector is standardized and shown. Is there any result of direct comparison between empty vector and mutant vector?
No significant differences were found between basal ACTH secretion in empty vector and mutant USP8 transfected cells. We added this information in the revised text (legend of Fig.2).
- L274- There are two periods at the end. The authors should also read the other texts carefully to see if there are any such mistakes. In addition, the authors should use the English proofreading service as needed.
As suggested by the reviewer, we have carefully revised the text and corrected the errors.
- L294(Fig3)- From (a) to (e), the characters on the horizontal axis are uneven and very difficult to see. Also, I'm worried that the letters of POMC are off center. Check all figures once for such mistakes.
We have modified all the figures as suggested by the reviewer.
- L313(Fig4)- What was “bas” ? (a) The FLAG-USP and GAPDH photos are the same as in Fig. 1, so they should be taken for only Fig. 4.
“Bas” is referred to control cells incubated without pasireotide. We have added this information in the revised figure legend. FLAG and GAPDH are the same as in Fig.1 because we used the same membranes for stripping and reprobing with the antibody for CD3. We have decided to insert these pictures above CD3 to allow a direct comparison of the intensity of the bands in each lane, together with the values of densitometric analysis showed below. The same consideration has been made for Fig.1, that shows SSTR5 together with FLAG-USP8 and GAPDH. However, upon reviewer’s request, in order to avoid to show the same pictures, we can add the CD3 picture in the Fig.1 and delete Fig4a.
Reviewer 2 Report
The authors have performed elegant and very exhaustive research on the implication of USP8 mutations in the behaviour of functioning corticotroph tumours, specifically on their sensitivity or insensitivity to the treatment with pasireotide. Although the translation of their results is, at the moment, low, they have made an essential contribution to the pathogenesis of functioning ACTH-secreting tumours.
Only minor questions:
- How many of the 7 ACTH-secreting 7PitNEts were lower than 9 mm?
- Were there any invasive, proliferative, or invasive/proliferative tumours?
- Was the impact of USP mutant on SSTR5 protein expression level distinct in small tumours than in larger ones?
- Results 3.1.
- Why did you study the impact of USP8 mutant on SSTR5 protein expression in only 3/5 USP8-wild type tumours (#1, #2 and #3)? What were the characteristics of the three chosen tumours concerning those not chosen (2)?
- Discussion
- “Moreover, we reported a significant increase in SSTR5 protein expression in AtT-20 cells transfected with the most frequently encountered USP8 mutants”. However, in results 3.1 you said, “Moreover, we reported a significant increase in SSTR5 protein expression in AtT-20 cells transfected with the most frequently encountered USP8 mutants with the exception of S718P USP8”. The last expression must be included in the discussion
Author Response
Reviewer 2
The authors have performed elegant and very exhaustive research on the implication of USP8 mutations in the behaviour of functioning corticotroph tumours, specifically on their sensitivity or insensitivity to the treatment with pasireotide. Although the translation of their results is, at the moment, low, they have made an essential contribution to the pathogenesis of functioning ACTH-secreting tumours.
We thank the reviewer for the careful reading of our manuscript and for her/his comments. All the issues raised have been addressed.
Only minor questions:
- How many of the 7 ACTH-secreting 7PitNEts were lower than 9 mm?
All the ACTH-secreting tumors were microadenomas smaller than 9 mm, two of them were not visualized at the MRI but only during surgery. We have added this information in the revised text (lines 108-110, Materials and Methods: “All the tumors were non invasive microadenomas with low ki67 (<3%), except for tumor #6 that had a ki67 of 4%. Two tumors were not visualized at the MRI but only during surgery”).
- Were there any invasive, proliferative, or invasive/proliferative tumours?
None of the 7 tumors used in this study were invasive. Ki67 were lower than 3% in all samples except sample #6 (ki67=4). We have added this information in the revised text (lines 108-110, Materials and Methods).
- Was the impact of USP mutant on SSTR5 protein expression level distinct in small tumours than in larger ones?
The impact of USP8 mutant on SSTR5 expression was tested in tumors #1,2 and 3. All these tumors were small, thus no comparison could be made.
- Results 3.1. Why did you study the impact of USP8 mutant on SSTR5 protein expression in only 3/5 USP8-wild type tumours (#1, #2 and #3)? What were the characteristics of the three chosen tumours concerning those not chosen (2)?
Unfortunately we could evaluate the expression of SSTR5 in only 3 tumors due to the low number of cells that can be obtained from corticotroph tumors. Indeed, in tumors #4 and #5 we could only perform the ACTH-secretion assay. The characteristics of the tumors have been added in the revised version of the manuscript (lines 108-110).
- Discussion. “Moreover, we reported a significant increase in SSTR5 protein expression in AtT-20 cells transfected with the most frequently encountered USP8 mutants”. However, in results 3.1 you said, “Moreover, we reported a significant increase in SSTR5 protein expression in AtT-20 cells transfected with the most frequently encountered USP8 mutants with the exception of S718P USP8”. The last expression must be included in the discussion
As suggested by the reviewer, we modified the sentence as follows: Moreover, we reported a significant increase in SSTR5 protein expression in AtT-20 cells transfected with the most frequently encountered USP8 mutants, with the exception of S718P USP8 (line 419).
Round 2
Reviewer 1 Report
The authors have revised this article, but it still has serious problems.
The authors explained that "basal does not present error bar since is the mean of different experiments in each of them the values of the treated groups have been expressed as a percentage of the respective basal." However, I cannot understand. Error bars are absolutely necessary if the means are normalized, even if they are different experiments. However, if all data were compared to basal each time, error bars would not be necessary, in which case it would be inappropriate to perform a t-test. The statistical tests should be described in each figure legend. Without a proper presentation of the data and the statistical tests, I cannnot fully judge the manuscript.
L52-59; The authors have responded appropriately to my concerns, providing additional references. If possible, it would be kind to the reader if you could write the percentages for just the top two, S718del and P720R.
Author Response
Reviewer 1
We thank the reviewer for the careful reading of our manuscript and for her/his comments. All the issues raised have been addressed.
The authors explained that "basal does not present error bar since is the mean of different experiments in each of them the values of the treated groups have been expressed as a percentage of the respective basal." However, I cannot understand. Error bars are absolutely necessary if the means are normalized, even if they are different experiments. However, if all data were compared to basal each time, error bars would not be necessary, in which case it would be inappropriate to perform a t-test. The statistical tests should be described in each figure legend. Without a proper presentation of the data and the statistical tests, I cannnot fully judge the manuscript.
To address the Reviewer’s concerns, all the statistical analyzes of this work have been revised by a professor of medical statistics. She has been added as a coauthor in the revised version of this manuscript (Prof. Francesca Bravi). We confirm that all data have been compared to basal each time. Indeed, results of densitometrical analysis of SST5 expression depend on several factors (such as exposure time, efficiency of primary and secondary antibodies, gel running and transfer to nitrocellulose membrane, ...) and the basal values differ greatly between one experiment and the other. You can find in the attached file the graph with the means±SD of the different experiments when the transfected cells were not normalized to the respective basal (fig.1b). However, we prefer to maintain the graph with normalized values. Moreover, we have modified the paragraph of statistical methods as follows: “The results are expressed as the mean ± S.D. A paired two-tailed Student’s t-test was used to detect the significance between two series of data. p < 0.05 was accepted as statistically significant.” Indeed, there were some mistakes in this paragraph, since methods not used in this work were mentioned. We used t test for all the figures, thus we did not add this information in each figure legend. However, we can add this information upon Reviewer’s request.
L52-59; The authors have responded appropriately to my concerns, providing additional references. If possible, it would be kind to the reader if you could write the percentages for just the top two, S718del and P720R.
We have added the percentage of the two mutants S718del and P720R (line 70-71): “…S718del representing the most frequent mutation (25.3%), followed by P720R (24.9%)”.

Round 3
Reviewer 1 Report
I understand that metrical analysis of SST5 expression depend on several factors (such as exposure time, efficiency of primary and secondary antibodies, gel running and transfer to nitrocellulose membrane, ...) and the basal values differ greatly between one experiment and the other. However, if all data are to be compared to the control each time, it is inappropriate to either display the data in a bar and error bars or to perform a t-test. It is advisable to consider again, for example, displaying the data as dot plots and using appropriate statistical analysis such as Wilcoxon's signed rank test.
You may refer to the following paper and others for how to present the results.
PLoS Biol. 2015 Apr 22;13(4):e1002128. doi: 10.1371/journal.pbio.1002128. Beyond bar and line graphs: time for a new data presentation paradigm.
Author Response
All the analyses have been revised by a professor of statistics (Bravi F). The statistical analysis used (t test) is appropriated. We agree with the reviewer that data can be presented as dot plots and analyzed with Wilcoxon's signed rank test, but we prefer to maintain the presentation with bar graph that is much clearer for the reader.
Round 4
Reviewer 1 Report
I will leave this matter to the Editor's discretion.
Author Response
We thank the Reviewer for all her/his comments and suggestions. We have modified the manuscript accordingly to Editor's comments.